

# Influence of shearing rate on the residual strength characteristic of
# three landslides soils in loess area
Baoqin Lian [1,2], Jianbing Peng[1], Xingang Wang[3*], Qiangbing Huang[1]
[1]College of Geological Engineering and Surveying, Chang'an University, Key Laboratory of Western China Mineral
Resources and Geological Engineering, Xi'an 710054, China
[2]Department of Geology & Geophysics, Texas A&M University, College Station, TX 77843-3115, United States
[3]State Key Laboratory of Continental Dynamics, Department of Geology, Northwest University, Xi'an 710069, China
*Corresponding author: Xingang Wang (xgwang@nwu.edu.cn); ORCID:0000-0002-1744-8712




**Abstract**
In order to investigate the effect of the shearing rate on the residual shear strength of slip zone soils, a series ring shear tests
were carried out on slip zone soils from three landslides in loess area at the two shearing rates (0.1mm/min and 1 mm/min).
The slip zone soil specimens used in present study were from the northwest of China. Results indicated that the shear
displacement to achieve the residual stage for specimens with higher shearing rate is greater than that of the lower rate.
Relationship between the residual friction coefficients and normal stress shows that the residual friction coefficients for all
specimens under the lower normal stress were greater than that under the higher normal stress at two shearing rate.
Furthermore, the difference in the residual friction angle $\phi_r$ at the two shearing rates, $\phi_r$ (1)- $\phi_r$ (0.1), under each normal
stress level were either positive or negative values, with the maximum absolute value of $\phi_r$ (1) - $\phi_r$ (0.1) reach up to 2.218°.
However, the difference $\phi_r$ (1) - $\phi_r$ (0.1) under all normal stresses was negative, which indicates that the residual shear
parameters reduced with the increasing of the shearing rate in loess area.



Keywords: Residual shear strength; Ring shear test; Shearing rate; Normal stress; Slip zone soils





## 1.  Introduction


Residual strength of soil is of great significance for evaluating the reactivating potential of the slope, in which consists of
pre-existing sliding surface. Residual strength of a landslide soil is defined as the minimum constant value of strength along
the slip plane, in which the soil particles are reoriented and subjected to sufficiently large displacements in relatively low
shearing rate (Skempton, 1985) .
Numerical studies have been done to assess the residual strength through the laboratory tests using ring shear tests and
reversal direct shear tests (Chen and Liu, 2013;Vithana et al., 2012). It is a generally accepted fact that the measurement of
the residual strength is most preferred done with a ring shear test since it allows the soil specimen be sheared at unlimited
displacement which can simulate the field conditions more accurately (Lupini et al., 1981;Tiwari and Marui, 2005;Bhat,
2013;Sassa et al., 2004). Until now, several relationships between the residual strength and soil index parameters have been
reported in the literature with a wide range of soil by using various kinds of ring shear apparatus (Hoyos et al., 2014;Jiang et
al., 2016;Kimura et al., 2015;Li et al., 2013;Skempton, 1964). Furthermore, many studies have shown that the shearing rate
may or may not affect the minimum value of soil strength at residual states (Suzuki et al., 2007;Grelle and Guadagno,
2010;Gonghui et al., 2010;Bhat, 2013;Tika and Hutchinson, 1999;Lemos, 1985;Morgenstern and Hungr, 1984;Tika, 1999).
From the high shearing rate aspect in the geotechnical literatures, Morgenstern and Hungr (1984) carried out ring shear
tests on two types of coarse sand in high velocity and found that the frictional behavior was not affected by either the
velocity or the normal stress. However, there were many researchers asserted that the effect of shearing rate on the shear
behavior of soil cannot be ignored. For example, Skempton (1985) , Tika and Hutchinson (1999) found that the faster
shearing rate above 100 mm/min may bring about great qualitative changes in the residual behavior. Moreover, Tika et al.
(1996) conducted fast ring shear tests on a wide range of natural soils and concluded that there are three types of rate effects
on the residual strength, namely, a positive rate effect (the residual strength of soil at fast rate is higher than that of the slow
rates), a neutral rate effect (the residual strength of soil is independent of the shearing rate) and a negative rate effect ( the
residual strength of the soil at higher speed is lower than that of the lower speed). Recently, Gratchev Ivan and Sassa (2015)
reported that the residual strength of the clay decrease with the shear rate increase from 0.2 to 5 mm/s.
On the other hand, in the slow shearing rate range, Skempton (1985) reported that variation in the value of the residual



friction angle for shearing rates in a range of 0.05 to 0.35 mm/min was less than a 5% and concluded that the impact of
shearing rate on the residual strength of clay is almost negligible within slow rate displacement. Similarly, Bhat (2013)
concluded that there is hardly increase in residual strength of kaolin clay with the shearing rate ranging from 0.233 mm/min
to 0.586 mm/min. Furthermore, Yokota et al. (1995) showed that residual strength is not affected by shearing rate lower than
1.01 mm/min in ring-shear tests. Except the above studies, other similar results were also found in clays that the residual
strength is independent of the shearing rate (Chen and Liu, 2013;Tiwari and Marui, 2001). However, Suzuki et al. (2001) has
reported the shearing rate ranging from 0.02 to 2.0 mm/ min significantly affected the residual strength of kaolin clay and
mud stone. Moreover, Gonghui et al. (2010) also has reported that the residual shear strength of the weathered serpentinite is
positively dependent on the shear rate in the slow rate.

68       On general, the effect of the shearing rate on the residual strength of the soil has not been sufficiently studied in high and

slow shearing rate range. Furthermore, except for a few studies, researchers have not widely reported the impact of the
shearing rate on the residual strength of loess soil in relatively lower shearing rate range from 0.1mm/min to 1 mm/min.
However, it should be noted that the residual strength parameters obtained from using different shearing rate may be adopted
to provide a guide for designing some precision engineering which require high accuracy of the design parameters, thus, the
effect of the shearing rate on the residual strength of soils should be fully understood to determine the parameters with high
reliability. In addition, residual strength of soil plays a key role in assessing the stability analysis and evaluating the
reactivation potential of landslides which consists of pre-existing slip plane surface. Therefore, accurate determination of the
residual strength parameters and their dependence on the shearing rate may affect the stability evaluation of landslides. Thus,
it is necessary to study the residual strength variation of loess in rate of shearing in order to have a good understanding of the
suitable approach for the residual strength measurement.

79       In this backdrop, the present study investigated the effect of the shearing rate on the residual strength of soil samples

obtained from three landslides in
loessic- developed areas at two different shearing rates (0.1mm/min and 1 mm/min) by using a ring shear apparatus. The
main objective of this study was to examine the change in the residual strength parameters of loess at different shearing rates
and their relationship with the normal stress in naturally drained ring shear tests.




## 2. **Geological setting of landslide sites**

Soil samples from three reactivated landslides in the northwest of China were selected for this study. Soil samples used for the ring shear tests and index measuring tests predominantly consist of loess deposits and were collected in a disturbed condition. For convenience, the names of landslide sites were abbreviated into Djg, Ydg, and Dbz. Figure 1 shows the study sites and some views of the landslides.

**Dingjiagou landslide**

The Djg landslide, located at the mouth of Dingjia Gully in Yan'an of China, is geologically composed of upper loess and lower sand shale in the Yan-chang formation. The dustpan-shaped landslide is inclined to the east, with its inclination 75.85°. The landslide is 350 m in width, 180 m in length, 70 m in elevation. The average thickness of slip mass is around 20 m, and the volume of landslide totaled approximately $105 \times 10^4$ m$^3$. The slip mass is mainly constituted by loess, whereas the sliding bed consists of sand shale in Yan-chang formation. The thickness of the sliding zone varied from 30 to 50 cm. The front lateral region of the main slide section of the Djg landslide, where the sampling was performed, was found to be silty clay.

**Yandonggou landslide**

The Ydg landslide is located in the Qiaogou town of Yan'an in Shaan xi province of China. The top and the toe altitude of the landslide are about 1165 m and 1110 m above the sea level, with the height difference between the toe and the top of landslide about 55 m. The slides have well-developed boundaries with the main sliding direction of 240° and slope angle of 30°. From the landslides profile, the sliding masses from top to bottom were classified by $Q_3$ loess, $Q_2$ loess and clay soil, respectively. Multiple landslide activities had occurred in this site, and the soil samples used in this study were collected from $Q_2$ loess stratum within the slide ranged from 4.5 to 18 m in height.

**Dabuzi landslide**

The Dbz landslide is located in the middle part of Shaanxi province (about 108°51'36" east longitude and 34°28'48" north latitude), China, which is a semi-arid zone dominated by loessic geology. In this region, the investigated site is classified as a typical loess tableland with quaternary stratum. The sedimentary losses in this area are grey yellow, and the exposure stratum





in this area has been divided into two stratigraphic units, namely, the upper late Pleistocene (Q₃) loess and the lower
mid-Pleistocene (Q₂) loess, of which the Q₃ loess is younger. The Q₃ loess is closest to the surface and is up to approximately
12 m thick, while the thickness of Q₂ loess may reach an upper limit of about 50 m (Leng et al., 2018). The loess in this area
have well-developed vertical joints (Sun et al., 2009). The travel distance and the maximum width of the slip mass are
roughly estimated to be 121.55 m and 133.46 m, respectively. The armchair-shaped landslide shows an apparent sliding
plane, with an area of approximately 15,660 m² and about 66.25 m maximum difference in elevation. The main direction of
this landslide is approximately 355°. The exposed slip zone in the side scarp of the landslide, where the sampling was done,
was found to be entirely in the Q₂ loess stratum of the Dbz landslide site. The thickness of narrow- band slip zone loess is
less than 1.0 cm, inclined at around 65° to the horizontal direction. Since the band of slip zone is thin, mix soils which consist
of the slip zone soils, the very thin upper and lower parts of the loess are mingled together and served as the representative
samples of the slip zone loess in this site.

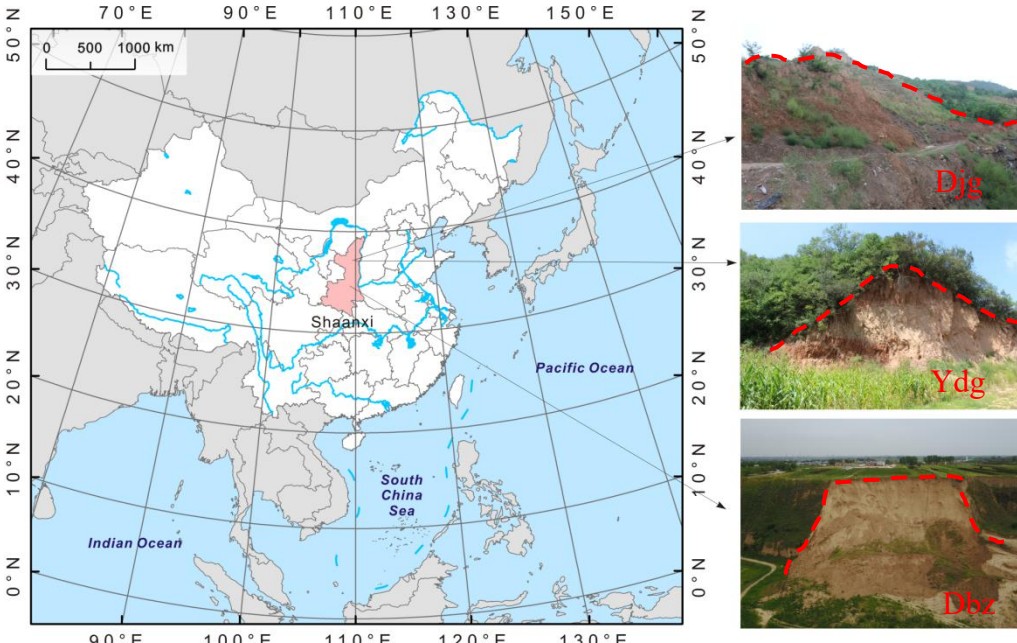

Figure 1. Location of study sites and some views of landslides
**3. Experimental scheme**
**3.1. Testing sample**



The fact that the residual shear strength is independent of the stress history was reported by many researchers (Bishop et
al., 1971;Vithana et al., 2012;Stark Timothy et al., 2005). Thus, disturbed loess samples from each of the three landslides
weighing about 25 kg were collected from the slip surface soil of each slide and used to determine residual shear strength.
The soil samples were air-dried and then crushed with a mortar and pestle, and subsequently processing it through 0.5 mm
sieve. Distilled water was added to the soil samples until desired density and water content were obtained. The physical
parameters such as natural moisture content, specific gravity, bulk density, plastic limit, and liquid limit were determined in
accordance with the Chinese National Standards   (CNS) GB/T50123-1999 (standards for soil test methods) (SAC, 1999),
but clay size was defined to be less than $2u$m followed ASTM, D422 (ASTM, 2007). Each soil sample was separated into
clay (sub 0.002 mm), silt (0.002-0.075 mm), and sand (0.075-0.5 mm) fractions. The physical indexes of the soil are listed in
Table 1.
The grain size distribution of soil was measured using a laser particle size analyzer Bettersize 2000 (Dandong Bettersize
Instruments Corporation, Dandong, China). The sieved soil samples were used to determine particle size distribution. In this
study, soil samples were treated with sodium hexaphosphate, serving as a dispersant, to disaggregate the bond between the
particles. The results show that the clay fraction in Djg landslide soil (24%) is more than two times than that from Ydg (9%)
and Dbz (9.1%). Furthermore, the particle size analyses illustrates that the percentage of silt-sized soil in three landslides
ranged from 75.66% to 87.4%. In addition, Ydg landslide soil consists of the greatest percentage of the sand fraction which
reaches up to 10.55%.
In present study, a total of twenty four specimens were tested at two shearing rate (0.1mm/min and 1 mm/min) and under
normal stresses ranged from 100kN/m² to 400kN/m² in a ring shear apparatus.
**Table 1** Physical parameters of slip-zone loess

| sites | $\rho_d$ | $W$ | $\rho$ | $G_S$ | $W_L$ | $W_p$ | Grain size fractions (%) | | | |
|---|---|---|---|---|---|---|---|---|---|---|
| | | | | | | | <0.002m | 0.002-0.005m | 0.005-0.075 | 0.075-0.5m |
| Djg | 1.74 | 19. | 2.0 | 2.6 | 3 | 20 | 24 | 11.48 | 64.18 | 0.34 |
| Ydg | 1.47 | 18 | 1.7 | 2.7 | 3 | 19 | 9 | 5.28 | 75.17 | 10.55 |



| Dbz | 1.48 | 16 | 1.7 | 2.7 | 3 | 21 | 9.1 | 6.4 | 81 | 3.5 |
|---|---|---|---|---|---|---|---|---|---|---|

Notes: $\rho_d$= dry density; w=moisture water content; $\rho$= bulk density; $G_S$ = specific gravity; $W_L$=liquid limit; $W_P$= plastic limit

**3.2.  Testing apparatus**

147        The advantage of a ring shear test apparatus to measure residual shear strength including its ability to allow

unidirectional shearing of a soil specimen (Bishop et al., 1971;Tika, 1999;Suzuki et al., 2007;Bromhead, 1979). Thus, a ring
shear apparatus was used in this study.

150        An advanced ring shearing apparatus (SRS-150) manufactured by GCTS (Arizona, USA) was adopted in ring shear

tests and the photos of apparatus were shown in Fig.2, which consists mainly of a shear box with an outer diameter of 150
mm, an inter diameter of 100 mm and the maximal sample height of 250 mm. The shearing box consists of the upper
shearing box and the lower shearing box. In the shearing process, the upper shearing box keeps still while the lower one
rotates. The apparatus, which provides effective specimen area of 98 cm$^2$, is capable of shearing the specimen for large
displacement in single direction. The annular specimen is confined by inside and outside metal rings. Moreover, the
specimen is confined by bottom annular porous plates and top annular porous plates in which have sharp-edged radial metal
fins which protrude vertically into the top and bottom of the specimen at the shearing process. The normal stress, shearing
strength and shearing displacement can be monitored while shearing by computer. The measurement features of the ring
shear apparatus employed in this study are described as follows: shearing rate range from 0.001 to 360 degrees per minute,
10 kN axial load capacity, 300 N.m continuous torque capacity, maximum normal stress of 1000 kN/m$^2$.





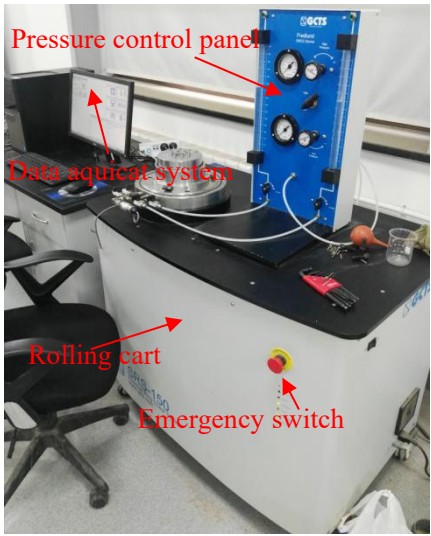
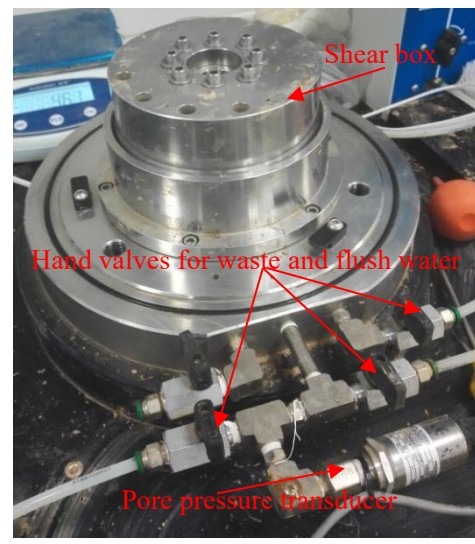


Figure 2. Ring shear apparatus (SRS-150)
**3.3.   Testing procedure**

164         In present study, reconstituted samples of the sub 0.5 mm soil fraction were used in the testing as it was reported that

the residual strength of the soil was unaffected by its initial structure (Bishop et al., 1971;Vithana et al., 2012). Specimens
were first prepared by adding distilled water to the air-dried soil until the saturated moisture contents of the three landslide
soils were obtained. Then, specimens were kept in a sealed container for at least one week to fully hydrate. Specimens are
then reconstituted in the ring-shaped chamber of the apparatus by compaction. In order to make the sample uniform while
packing, the sample was placed in three layers, and each layer was tamped under a vertical stress which is lower than the
given normal stress to achieve the design height. The final height of the specimen in the ring shear apparatus after tamp
varied but was typically about 20 mm (to achieve a specific bulk density). The specimen was then consolidated under a
specific effective normal stress in a range of 100kN/m$^2$ to 400kN/m$^2$ until required consolidation was achieved. Then, the
consolidated specimen is subjected to shearing under constant normal stress by rotating the lower half of the shear box
attached to a gear, while the upper half remains still. In ring shear tests, the normal stress at the shearing was the same as at
consolidation stage.

176         In this study, ring shear tests were performed in a single stage under drained condition and the samples were subjected

to shear until the residual state was achieved. Drain condition of the shearing process is provided by two porous stones



attached on the top and the bottom platen of the specimen container. As for soil specimens with low permeability, the rate of
excess pore pressure generation in the shear box may exceeded that of pore-pressure dissipation, this type of condition is
identified as naturally drained condition in previous studies(Okada et al., 2004). Furthermore, Tiwari (2000) asserted that it
was acceptable to use a shearing rate below 1.1 mm/min to simulate the field naturally drained condition. Thus, shearing
rates of 0.1 mm/min and 1 mm/min were used in this study to simulate the naturally drained condition of the slip zone soils.
**4.    Results and discussions**
Twenty four specimens were tested to investigate the shear characteristics of the slip-zone soils in the ring shear
apparatus. Tests results are shown in this section.
**4.1. Shear behavior**
Figures 3a, 4a and 5a show the typical shear characteristics of the slip-zone soils (shearing rate 0.1 mm/min and 1
mm/min) obtained from three different locations, where, the shear stress is plotted against the shear displacement at the
normal stress ranged from 100kN/m$^2$ to 400kN/m$^2$. It is a widely accepted fact that normal stress has effect on the shear
behavior of the soil (Kimura et al., 2015;Stark Timothy et al., 2005;Stark et al., 2005;Eid, 2014), thus, the shear behavior of
samples at the peak and residual stages, where, the determined peak friction coefficient as well as residual friction coefficient
are plotted in Figure 3b, 4b, and 5b against the corresponding effective normal stresses. The friction coefficient is defined as
the shear stress divided by the effective normal stress.
Figures 3a, 4a and 5a demonstrate that shear stress increases dramatically within small shear displacement and then
reduces with shearing displacement, until residual conditions were achieved at large displacements. Furthermore, it is clear
that peak strength as well as residual strength of the samples with high shearing rate is almost smaller than that of the
samples with low rate. In Figures 3a, 4a and 5a, a clear drop can be seen, at any normal stress, for specimens obtained from
all sites. It is obvious that Djg specimens showed greater peak-post drop than that of Ydg and Dbz specimens. According to
the conclusion that the residual stage is attained if a constant shear stress is measured for more than half an hour (Bromhead,
1992), it can be seen    that the shear displacement to achieve the residual stage for specimens with higher shearing rate is
greater than that of the lower rate. For example, the minimum shear displacements for attaining residual condition for Djg
specimens with low and high shearing rate were about 360mm and 650mm, respectively. Under the shearing rate of



0.1mm/min and 1mm/min, Ydg specimens need approximately 80mm and 1,400 mm displacement to achieve residual stage.
However, Dbz specimens require about 40mm and 60mm displacement to reach residual condition for low and high shearing
rate, respectively.

**4.2. Effect of normal stress on the friction coefficients**
It can be seen from the Figures 3b, 4b and 5b that the friction coefficients (peak and residual) are higher at lower
effective normal stress levels. For example, the peak and residual friction coefficient of Djg landslide soils at the shearing
rate of 0.1mm/min reduced from 0.569 to 0.32 and from 0.3 to 0.262, respectively. Similarly, results obtained from other two
landslides soils also showed that the friction coefficients decrease nonlinearly with the normal stresses. Furthermore,
specimens with shearing rate of 0.1mm/min attained greater friction coefficients than that with shearing rate of 1mm/min.
In order to get an insight into the effects of the normal stress on the slip zone shear strength, the shear behavior of the
soil sheared at the normal stress of $100kN/m^2$ and $400kN/m^2$ were selected for analysis. At the normal stress of $100kN/m^2$,
Djg samples showed about 47.3% and 36.8% decrease in the friction coefficient from the peak friction coefficient at the
shearing rate of 0.1 and 1 mm/min, respectively, which is greater than in the Ydg (about 9.8% and 10.3%) and Dbz (about
2.4% and 3.2%) samples. In Figures 3b, 4b and 5b, on average, it is obvious that the decrease of the friction coefficient from
the peak strength to the residual strength in the Djg sample was almost 18.1% and 21.3% for the sample consolidated at
normal stress of $400kN/m^2$ under the shearing rate of 0.1mm/min and 1mm/min (Figure 3b), While the friction coefficient
reduction in Ydg sample with low and high shearing rate were only about 4.1% and 4.8% (Figure 4b). And the friction
coefficient reduction in Dbz samples with low and high rate were only approximately 5.6% and 6.0% (Figure 5b) from the
peak strength, respectively. Based on the conclusion that the post-peak drop in strength of soil is only due to particle
reorientation after the peak strength (Mesri and Shahien, 2003;Skepmton, 1964), the results in this study demonstrated that
the Djg landslide soil existed the greater particle reorientation compared with that of other two landslide soils.




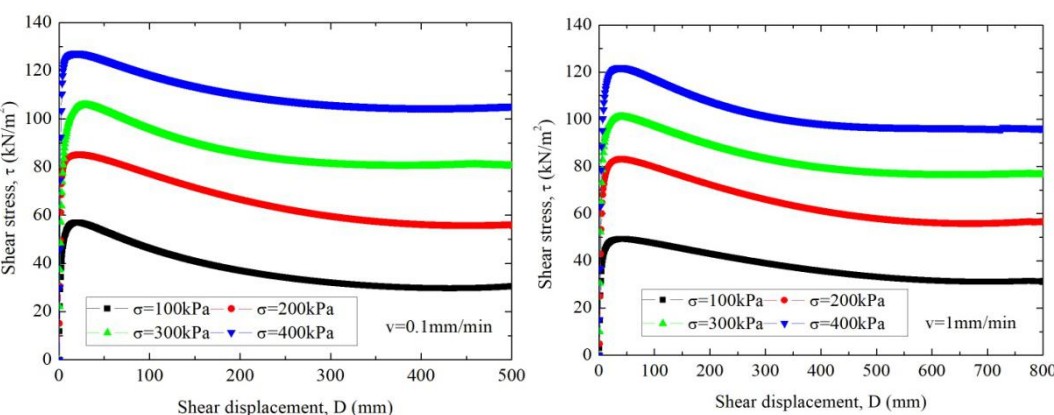

(a)Relationship between shear stress and shear displacement

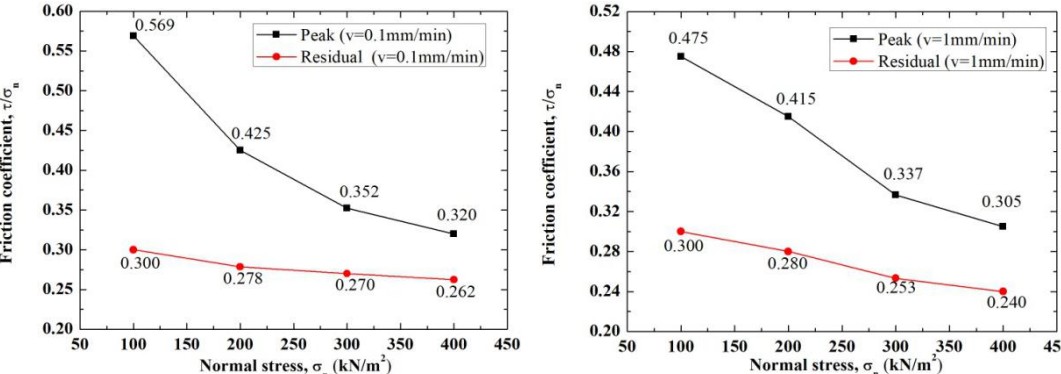

(b)Relationship between friction coefficient and normal stress

Figure 3. Shear behavior characteristics of Djg soil samples

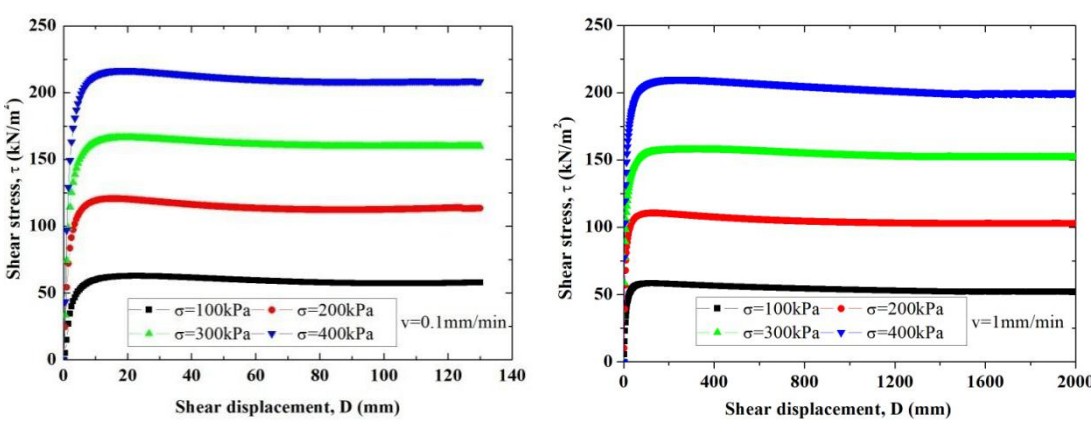

(a)Relationship between shear stress and shear displacement




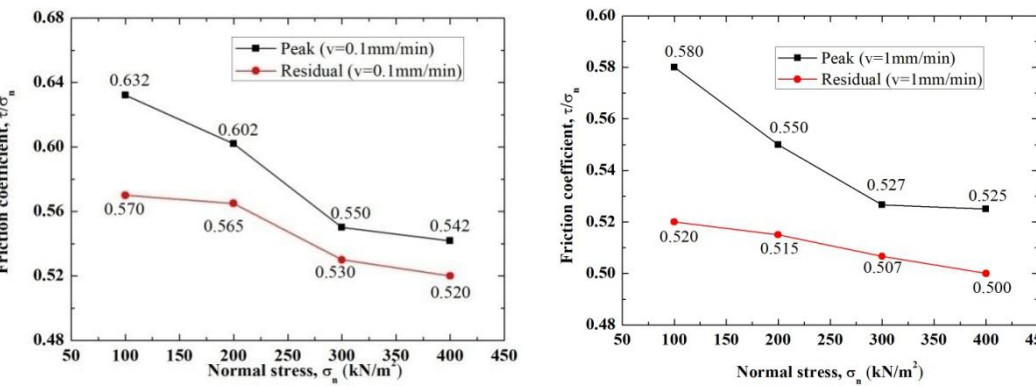

(b) Relationship between friction coefficient and normal stress
Figure 4. Shear behavior characteristics of Ydg soil samples

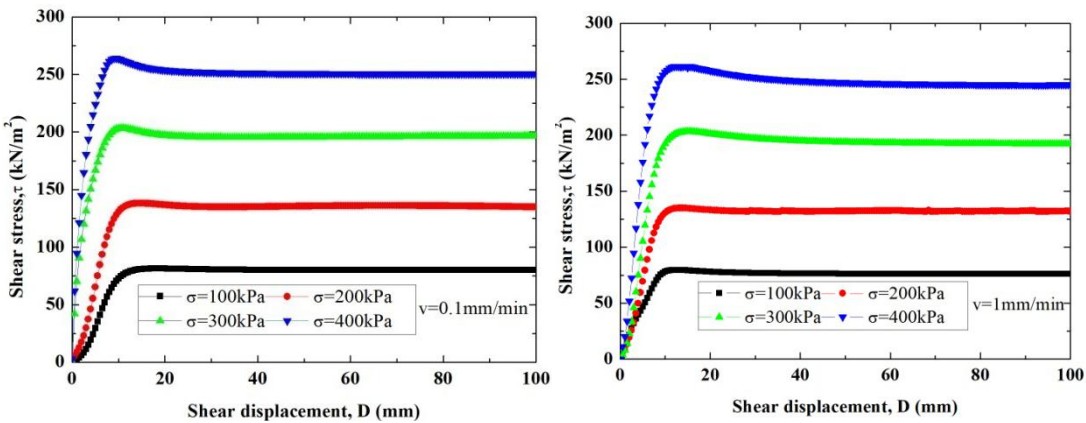

(a) Relationship between shear stress and shear displacement

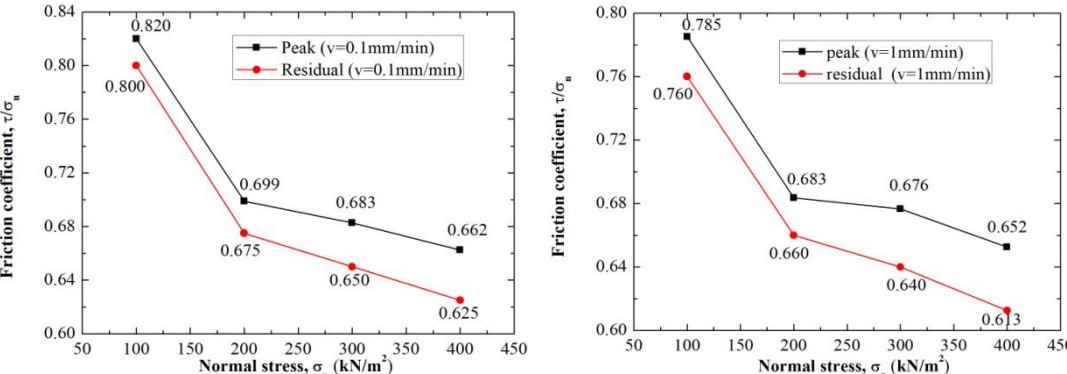

(b)    Relationship between friction coefficient and normal stress
Figure 5. Shear behavior characteristics of the Dbz soil samples





### 4.3. Effects of shearing rate on residual strength parameter


For the representative samples described above, Figures 6, 7 and 8 show the relationships between the residual friction
coefficient and the normal stress, and the residual strength parameters. The residual friction coefficient is plotted against the
normal stress. The residual friction coefficient is defined as the residual shear strength divided by normal stress. It is widely
recognized that the shear strength parameters including cohesion and friction angle (Terzaghi, 1951;Stark et al., 2005).
However, according to the previous studies, the residual angle of soils varies depended on the soil properties as well as the
magnitude of normal stress provided the residual cohesion of soil is zero (Skepmton, 1964;Bishop, 1971;Kimura et al., 2014).
Thus, in this study, the residual frictions are calculated by Coulomb's law assumed the residual cohesion is zero. The
residual strength parameters were defined as $\phi_r(0.1)$ and $\phi_r(1)$ at the low shearing rate and high shearing rate, respectively.
And the difference between the residual friction angles at two shearing rate was defined as $\phi_r(1)$ - $\phi_r(0.1)$. Comparatively,
the residual friction coefficient was defined as $\tau_r/\sigma_n(0.1)$ at the low shearing rate and $\tau_r/\sigma_n(1)$ at the high shearing rate,
respectively. Furthermore, the difference between the residual friction coefficients was defined as $\tau_r/\sigma_n(1)$ - $\tau_r/\sigma_n(0.1)$ . Table
2 summarized the residual parameters of the landslide soils.
Figure 6 shows that the residual friction coefficients were relatively low in Djg samples. The coefficients $\tau_r/\sigma_n(0.1)$
and $\tau_r/\sigma_n(1)$ at the normal stress of 100kN/m² to 400kN/m² ranged from 0.3 to 0.262 and from 0.3 to 0.24,respectively. The
difference between the friction coefficients, $\tau_r/\sigma_n(1)$-$\tau_r/\sigma_n(0.1)$, at each normal stress level are varied in a range of -0.022 to
+0.002. For the difference between the residual friction angles, $\phi_r(1)$- $\phi_r(0.1)$, ranged from -1.212° to +0.079° (Table 2). For
normal stress above 200kN/m², the coefficient $\tau_r/\sigma_n(0.1)$ was found to be greater in the magnitude than the coefficient $\tau_r/\sigma_n(1)$.

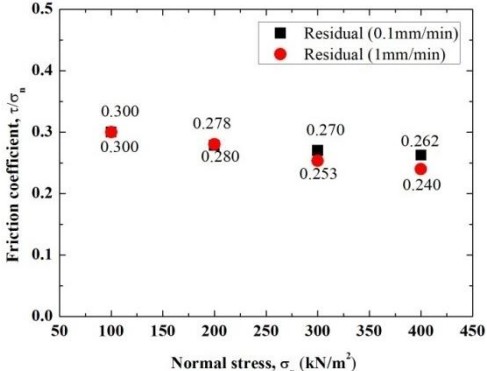

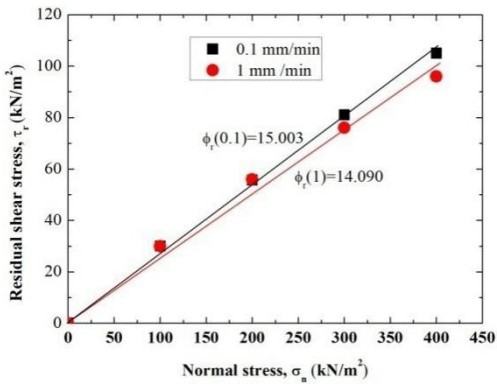



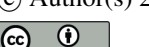

Figure 6. Relationships between residual shear stress and normal stress, and residual strength parameter for Djg soil
sample

Figure 7 gives the relationship between the residual coefficient and normal stress, and residual shear strength parameter
for Ydg samples. The coefficients $\tau_r/\sigma_n$ (0.1) and $\tau_r/\sigma_n$ (1) under the normal stress of 100kN/m$^2$ to 400kN/m$^2$ ranged from 0.57
to 0.52 and from 0.52 to 0.50, respectively. Furthermore, the difference $\tau_r/\sigma_n$ (1)-$\tau_r/\sigma_n$ (0.1) at each normal stress was from
-0.05 to -0.02. As for the difference between the residual friction angles, $\phi_r$ (1) - $\phi_r$ (0.1), was in a range of -2.218° to -0.909°.
In case of Ydg soil sample, there was insignificant reduction in residual friction coefficients with the increasing of shearing
rate for all normal stresses.

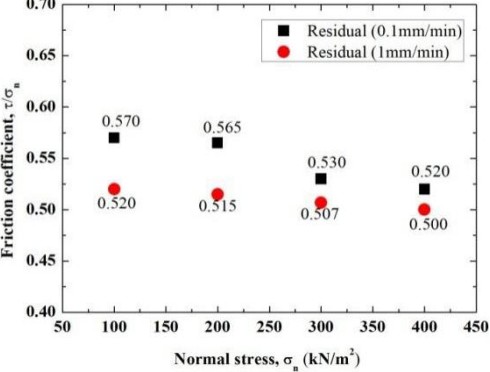
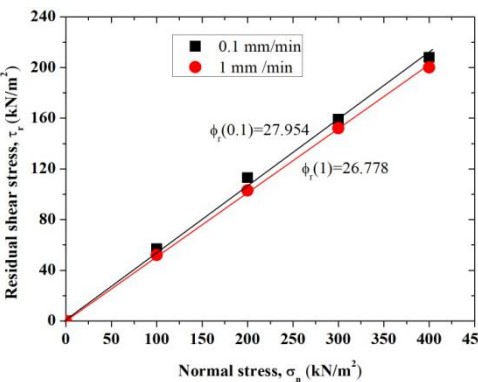

Figure 7. Relationships between residual shear stress and normal stress, and residual strength parameter for Ydg soil sample

Figure 8 presents the results of the Dbz samples. The coefficients $\tau_r/\sigma_n$ (0.1) and $\tau_r/\sigma_n$ (1) at the normal stress of 100kN/m$^2$ to
400kN/m$^2$ ranged from 0.8 to 0.625 and from 0.76 to 0.613, respectively. The difference $\tau_r/\sigma_n$ (1)-$\tau_r/\sigma_n$ (0.1) at each normal
stress was from -0.04 to -0.01. The difference $\phi_r$(1)- $\phi_r$(0.1) was from -1.425°  to -0.405°. For Dbz samples, there was
somewhat decrease of the residual friction coefficients with the increasing of the shearing rate for all normal stress levels. It
is noted that the maximum difference was found at the lowest normal stress of 100kN/m$^2$.
Table 2 summarizes residual strength parameters including $\phi_r$ (0.1) and $\phi_r$ (1) of all specimens obtained from the ring
shear tests in this study. As for the Djg samples, the residual strength parameter $\phi_r$(0.1) and $\phi_r$(1) for all normal stress were

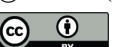


found to be 15.003° and 14.09°, respectively. However, the residual friction angles $\phi_r(0.1)$ and $\phi_r(1)$ of the Ydg samples
were obtained to be 27.954 ° and 26.778°, respectively. In the case of Dbz sample, the friction angles $\phi_r(0.1)$ and $\phi_r(1)$ were
high, 32.822° and 32.293°, respectively. The residual friction angles $\phi_r(0.1)$ and $\phi_r(1)$ under all normal stresses were from
15.003° to 32.822° and from 14.09° to 32.293°, respectively.
Due to the influence of the shearing rate, the difference $\phi_r(1) - \phi_r(0.1)$ in the Djg, Ydg and Dbz samples, were -0.913°,
-1.176° and -0.529° , respectively. Wang (2014) and Fan et al. (2017) asserted that the residual shear strength of remoulded
loess hardly affected by shearing rate below 5mm/min. However, the results in this study shown that $\phi_r(1) - \phi_r(0.1)$ under
all normal stress levels were negative for slip zone loess. Moreover, the maximum value of the difference $\phi_r(1) - \phi_r(0.1)$ even
reached about 1.176°.

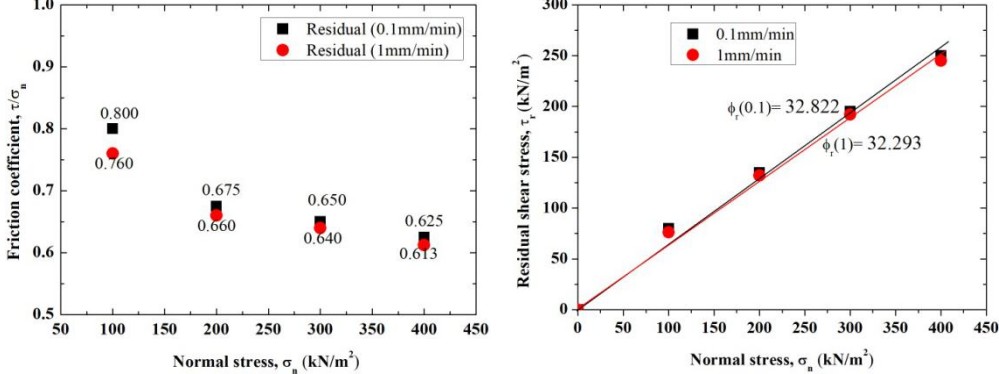


Figure 8. Relationships between residual shear stress and normal stress, and residual strength parameter for Dbz soil sample

Table 2 Residual shear strength parameter of landslide soils

| No | Sample | Normal stress(kN/m²) | Residual strength parameter | | | | Difference in parameter | |
|---|---|---|---|---|---|---|---|---|
| | | | 0.1 mm/min $\phi_{r\,(0.1)}$ ($c_{r(0.1)}$=0) (Degrees) | | 1 mm/min $\phi_{r(1)}$ ($c_{r(1)}$=0) (Degrees) | | $\phi_{r(1)}$- $\phi_{r(0.1)}$ (Degrees) | |
| | | | Under each $\sigma_n$ | Under all $\sigma_n$ | Under each $\sigma_n$ | Under all $\sigma_n$ | Under each $\sigma_n$ | Under all $\sigma_n$ |





| 1 | Djg | 100 | 16.699 | 15.003 | 16.699 | 14.090 | 0 | -0.913 |
|---|-----|-----|--------|--------|--------|--------|------|--------|
|   |     | 200 | 15.563 |        | 15.642 |        | 0.079 |        |
|   |     | 300 | 15.110 |        | 14.216 |        | -0.894 |       |
|   |     | 400 | 14.708 |        | 13.496 |        | -1.212 |       |
| 2 | Ydg | 100 | 29.683 | 27.954 | 27.474 | 26.778 | -2.209 | -1.176 |
|   |     | 200 | 29.466 |        | 27.248 |        | -2.218 |       |
|   |     | 300 | 27.923 |        | 26.870 |        | -1.053 |       |
|   |     | 400 | 27.474 |        | 26.565 |        | -0.909 |       |
| 3 | Dbz | 100 | 38.660 | 32.822 | 37.235 | 32.293 | -1.425 | -0.529 |
|   |     | 200 | 34.019 |        | 33.425 |        | -0.594 |       |
|   |     | 300 | 33.024 |        | 32.619 |        | -0.405 |       |
|   |     | 400 | 32.005 |        | 31.487 |        | -0.518 |       |


## 5. Influence of the shearing rate on the residual friction angles according to soil properties


Figure 9 depicts the relationships between residual friction angles as well as the difference in the residual friction

angles and soil properties including liquid limit (LL), plasticity index (Ip) and clay fraction (CF) at two shearing rates.

The residual friction angles at two shearing rates decreased nonlinearly with the increasing of the LL. As for the

relationship between the $\phi_r$ and Ip, the $\phi_r$ under the low and high shearing rates decreases from about $32°$ to $15°$ with

increasing the plasticity index from 11 to 16. With increasing of CF from 9% to 24%, the residual friction angles under

low and high shearing rates were found to decrease. Interestingly, for Dbz and Ydg soils of which have similar

percentage of clay fraction, the residual friction angles at both shearing rates varied. However, in the relationships

between the difference in the residual friction angles and the soil properties, no clear correlations were found.





Figure 9. Relationships between residual shear parameter, the difference in residual shear parameter and the soil

properties at two shearing rates



**Conclusions**

The shearing rate of slip zone soil of landslide may be changed after the occurrence of the first landslide activity, thus, the residual shear strength of the slip zone soil could be changed accompanying this process. As for some precision engineering which require high accuracy of the design parameters, the shearing rate effect on the residual shear strength should be fully investigated. In this study, at the shearing rate of 0.1 mm/min and 1 mm/min, a series of ring shear tests under normal stress ranged from 100kN/m$^2$ to 400kN/m$^2$ were performed on reconstituted slip zone soil samples obtained from three landslides in loess area. The main results can be summarized in the following points:

(i)  The shear displacement to achieve the residual stage for specimens with higher shearing rate is greater than that of the lower rate.

(ii)  As for slip zone soils in this study, specimens with lower shearing rate attained greater friction coefficients than that with higher shearing rate.

(iii)  At the two shearing rate (0.1 mm/min and 1 mm/min), the residual friction coefficient under the lower normal stress was higher than that under the higher normal stress in all samples. In addition, there was a nonlinearly decrease trend of the residual friction with the normal stress.

(iv)  For slip zone soils in this study, the difference at the two shearing rate, $\phi_r(1) - \phi_r(0.1)$, under each normal stress level were either negative or positive. However, under all normal stress, the difference at the two shearing rate $\phi_r(1) - \phi_r(0.1)$ was found to be positive.

(v)  The residual friction angles reduce with the increasing of shearing rate. Furthermore, the maximum magnitude of the difference between the residual friction angle $\phi_r(1)$ and $\phi_r(0.1)$ was even obtained to be approximately 1.176° in loess area.

(vi)  The relationships between the $\phi_r$ under two shearing rates and soil properties including liquid limit and plasticity index, demonstrated that the $\phi_r$ at both shearing rates decrease gradually with the increasing of LL and Ip. However, no clear correlations between the difference in the $\phi_r$ at low and high shearing rates and the soil properties were found.



337

**Acknowledgments**

Financial support of National Key Fundamental Research Program of China (973) (Grant No.2014CB744700) and the Major

Program of National Natural Science Foundation of China (Grant No. 41790440) are gratefully acknowledged. The support

provided by China Scholarship Council (CSC file No. 201706560016) during a visit of the first author (Lian) to Texas A&M

University is sincerely acknowledged.

343



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
