# Peer review of "Influence of shearing rate on the residual strength characteristic of three landslides soils in loess area Baoqin Lian 1,2, Jianbing Peng1, Xingang Wang3\*, Qiangbing Huang1 1College of Geological Engineering and Surveying, Chang'an University, Key Laboratory of Western China Mineral Resources and Geological Engineering, Xi'an 710054, China"

_Natural Hazards and Earth System Sciences, 2018_

## Referee Comment (RC1) · Anonymous Referee #1 · 27 Sep 2018

Residual strength of soils is important for the evaluation of the reactivation potential of ancient landslides. Determination of residual strength of soils is challenging in geotechnical engineering, because, as well known, the residual strength is related to many factors, such as soil type, clay contents, plastic limit, moisture, slaking property, dynamic loading and rate of shear. This study performs ring shear tests on sliding surface soils collected from three landslides in loess area. Two rates of shear are considered to investigate the influence of rate of shear on the residual strength. This study matches the scope of the journal. The following comments should be carefully addressed and revision is required.

[Figure]

1) To the best of my knowledge, the influence of rate of shear on the soil residual strength has been widely reported in previous. My question is: what is the difference between the findings revealed in this study and those reported in previous? Where is the innovation of this study?

2) In Line 128, distilled water is added to the soil until the desired water content is achieved. but in Line 166, distilled water is added to soil until the saturated moisture is obtained. Which one is correct? Please check.

3) In Section 3.1, two rates of shear are selected to investigate the influence of the rate. Why are the two rates preferred? Please explain it.

4) As is known, shear behavior of soils is closely related to moisture. But in Section 3.3, samples only with saturated moisture are used. Please explain it.

5) Previous literature has revealed the shear stress fluctuates as shear displacement increases, making it difficult to determine exact value of residual shear strength. In this study, how to determine this value?

6) In Section 4.3, the Coulomb law is adopted to determine the residual frictions. This study states that, the law assumes that residual cohesion is zero. However, most previous works seem not require the assumption.

7) Errors are found in writing. 7a) The "a series" in Line 20 should be "a series of ". 7b) The "0.2"in Line 57 should be "0.2 mm/s". 7c) The "0.05" in Line 59 should be "0.05mm/min". 7d) The "a 5%"in Line 59 should be "5%". 7e) The "ring-shear tests" in Line 63 should be "ring shear tests". 7f) The "0.02" in Line 65 should be "0.02 mm/min". 7g) The "mud stone" in Line 66 should be "mudstone". 7h) The "range from" in Line 70 should be "ranging from". 7i) The "4.5" in Line 104 should be "4.5m". 7j) The "0.001" in Line 159 should be "0.001 degree". 7k) The "drain" in Line 177 should be "drained". 7l) The "Figure 3b, 4b, and 5b" in Line 192 should be "Figures 3b, 4b, and 5b". 7m) The "as$\tau$r/$\sigma$n (0.1)" and "and$\tau$r/$\sigma$n (1)" in Line 251 should be "as $\tau$r/$\sigma$n (0.1)" and "and $\tau$r/$\sigma$n

(1)". 7n) The "ÑĎr(1) - ÑĎr(0.1)" throughout the manuscript should be italic.

8) Errors are found in figures. In Figures 3, 4 and 5, the unit of normal stress should be "kN/m2". For example, in the Figure 4a, the unit of $\sigma$ should be "kN/m2" to keep consistent with the text body.

9) Apart from rate of shear, additional factors, such as slaking property and dynamic loading, also affect the residual strength of sliding surface of landslides, and in turn governs the stability of colluvial landslides. The following literature may be related to the additional factors and colluvial landsides, and may be cited, if applicable.

Li C, Yan J, Wu J, et al. Determination of the embedded length of stabilizing piles in colluvial landslides with upper hard and lower weak bedrock based on the deformation control principle. Bulletin of Engineering Geology and the Environment, 2017: 1-20.

Yan C, Xu X, & Huang L. Identifying the impact factors of the dynamic strength of mudded intercalations during cyclic loading. Advances in Civil Engineering, 2018.

Shen P, Tang H, Huang L, & Wang D. Experimental study of slaking properties of red-bed mudstones from the Three Gorges Reservoir area. Marine Georesources & Geotechnology (in the press).

---

## Referee Comment (RC2) · Anonymous Referee #2 · 18 Oct 2018

Review of manuscript nhess-2018-270 Title: "Influence of shearing rate on the residual strength characteristic of three landslides soils in loess area" by Baoqin et al.

Dear editor,

The paper presents the results of shear experiments conducted on 24 loess samples collected from three landslides. Samples were sheared in a rotary apparatus under two imposed velocities and four different normal stresses. The authors summarized the results and briefly discuss them. This manuscript should be better organized and written. Language mistakes make it sometimes impossible to read or understand. In addition, the authors should clarify what is innovative in this paper or how their results

support or differ from previous studies. I recommend the authors to significantly revise the manuscript before resubmission. Hope this helps

Major comments

1. I found it hard to understand the connection between the definition of the residual strength of a landslide and what is actually measured in the lab experiments. Some of my misunderstanding may cause by the fact that I'm not a soil mechanics expert. Saying that, I think that the authors should carefully define the basic terms and explain how they connect to their experiments. For example, the first sentence of the introduction says "Residual strength of soil is of great significance for evaluating the reactivating potential of the slope, in which consists of pre-existing sliding surface". I'm not sure how the current experiments deals with the "reactivating potential" as they suggest only one continuous sliding event in their experiments. In addition, how about the sliding surfaces? Did they form during the experiments?

2. It seems that one important point of the experiments is the imposed shear velocity. The authors should clearly define in the introduction what are low and high velocities. I would rather use m/s as the velocity units, but in any case they should be consistent along the paper (see lines 52 and 57 as an example).

3. The motivation for this study is not clear to me. In lines 68-69 the authors suggest that not enough studies conducted on the issue that they just summarized in details in the two paragraphs above it. They should clarify what is the gap and how their new study contributes to the understanding of the problem.

4. In the "Results and discussion" chapter many of the results are not discussed. For example, in sub-section 4.1 no discussion is following the observed difference between samples. Moreover, there are a lot of details with almost no discussion!

5. I really recommend the authors to send the manuscript to English proofreading before resubmitting the manuscript.
Moderate and minor comments

1. Figure 1 - Please increase line width of the arrows that point on "locations of study". I think it is better to add a small symbol for the location of each site. What are the red dashed lines in the photos? It should be described in the figure caption.

2. Line 127 – Why did you crush the material? Is it because the fragments are too big for the cell? Please explain in the text.

3. Line 128 - What is the desired content? Please refer to a table and/or give the numbers here.

4. Line 131 - When during the procedure you sieved the material? Before or after adding the distilled water? Before or after crushing?

5. Line 132 – "physical indexes". Do you mean the physical parameters that are listed in the line above?

6. Table 1 – Check units! Replace m with mm.

7. Lines 147-148 - It is not sounds to me like the most important thing about ring shear test apparatus, as direct shear is better in that sense. I would say that unlimited displacement is the most important advantage of the rotary shear for this specific application.

8. Line 155 – "single direction", what do you mean? It is a rotary shear and not direct shear apparatus!

9. Lines 155-157 and 178 – The annular porous plates, explain why they are needed! I recommend including an image that shows them.

10. Line 158 – What is the sampling rate of the rheology data?

11. Figure 2 - Please change the font color and add a background to improve visibility. I would highly recommend adding a sub-figure here that shows a cross section of the

shear box and clarify where the actual soil sample is, and which part is rotating and which is stationary.

12. Line 172 – What is the required consolidation? How do you know if you get it or not?

13. I would combine sections 3.1 and 3.3 or better separate the text.

14. Line 195 – The authors suggest they ran the samples for large displacements. Did they run at least one experiment beyond the maximum displacement given in the figures to ensure that they really get to a steady state friction.

15. Lines 222-224 - You should better explain what actually this reorientation is and why do you think it should be different specifically for Djg samples?

16. Line 296 - I'm not sure what the connection between these soil properties is. Please define them and better connect to the measurements.

17. The current Conclusions chapter is mostly a summary of the results, instead of a short description of what we have studied and can take from this experimental study.

---

## Author Comment (AC1) · 18 Dec 2018

Residual strength of soils is important for the evaluation of the reactivation potential of ancient landslides. Determination of residual strength of soils is challenging in geotechnical engineering, because, as well known, the residual strength is related to many factors, such as soil type, clay contents, plastic limit, moisture, slaking property, dynamic loading and rate of shear. This study performs ring shear tests on sliding surface soils collected from three landslides in loess area. Two rates of shear are considered to

investigate the influence of rate of shear on the residual strength. This study matches the scope of the journal. The following comments should be carefully addressed and revision is required.

We thank for your constructive comments. The manuscript has been significantly improved by incorporating your suggestions. The following are our point-to-point responses to your comments.

1)   To the best of my knowledge, the influence of rate of shear on the soil residual strength has been widely reported in previous. My question is: what is the difference between the findings revealed in this study and those reported in previous? Where is the innovation of this study?

Reply: With regards to the difference between the findings revealed in this study and those reported in previous: Due to the influence of the shearing rate, the difference

$\phi_r$ (1) - $\phi_r$ (0.1) in the Djg, Ydg and Dbz samples, were -0.913°, -1.176° and -0.529° , respectively. Wang (2014) and Fan et al. (2017) asserted that the residual shear strength of remolded loess hardly affected by shearing rate below 5 mm/min. However, the results in this study shown that $\phi_r$ (1) - $\phi_r$ (0.1) under all normal stress levels were negative for slip zone loess. Moreover, the maximum value of the difference $\phi_r$ (1)- $\phi_r$ (0.1) even reached about 1.176°. see details on lines 401-407 of the revised manuscript.

[Figure]

With regards to the innovation of this study: On general, the effect of the shearing rate on the residual strength of the soil has not been sufficiently studied in high and slow shearing rate range. Many studies above have been conducted on the residual shear strength of soils, and some inconsistent or even opposite results have been reported in the literature, which implied that there is still a lack of experimental data on this topic. Furthermore, research on the impact of the shearing rate on the residual strength of loess soil in relatively lower shearing rate range from 0.1 mm/min to 1 mm/min is scarce. See details on lines 96-102.

iscussion paper

2)    In Line 128, distilled water is added to the soil until the desired water content is achieved. but in Line 166, distilled water is added to soil until the saturated moisture is obtained. Which one is correct? Please check.

Reply:  In our study, all ring shear tests were conducted on saturated specimens. We have checked these two sentences and we have replaced "desired water content" with "saturated water content" in the revised manuscript, see line 183 of revised manuscript.

3)    In Section 3.1, two rates of shear are selected to investigate the influence of the rate. Why are the two rates preferred? Please explain it.

Reply:  Following the previous study conducted by Tika et al. (1996), shearing rate higher than 10 mm/min is defined as high shearing rate, whereas, shearing rate lower than 10 mm/min is defined as low shearing rate. See detail on lines 65-66 of revised manuscript.

In slow shear rate range, Yokota et al. (1995) showed that residual strength is not affected by shearing rate lower than 1.01 mm/min in ring-shear tests. However, Suzuki et al. (2001) has reported the shearing rate ranging from 0.02 to 2.0 mm/ min significantly affected the residual strength of kaolin clay and mud stone. Furthermore, research on the impact of the shearing rate on the residual strength of loess soil in relatively lower shearing rate is scare, thus, we chosen two shearing rates 0.1 mm/min

and 1 mm/min in slow shearing rate range to investigate the shearing rate influence on the loess under the naturally drained condition.

4) As is known, shear behavior of soils is closely related to moisture. But in Section 3.3, samples only with saturated moisture are used. Please explain it.

Reply: Yes, shear behavior of soils is closely related to moisture content of soils. Since loess is sensitive to changing in moisture content, its strength would dramatically reduce with increasing of moisture content. Therefore, in this study, we only focus on the saturated specimen corresponding to the lowest strength of loess.

5) Previous literature has revealed the shear stress fluctuates as shear displacement increases, making it difficult to determine exact value of residual shear strength. In this study, how to determine this value?

Reply: Actually, previous literature has revealed the shear stress fluctuates as shear displacements increases especially at high shear rates (Li ,2013). Following the research conducted by Bromhead (1992), the residual stage is attained if a constant shear stress is measured for more than half an hour in this study.

[Figure]

6)  In Section 4.3, the Coulomb law is adopted to determine the residual frictions. This study states that, the law assumes that residual cohesion is zero. However, most previous works seem not require the assumption.

Reply: Based on previous research (Bishop, 1971; Kimura et al., 2014; Skepmton, 1964), we assume the residual cohesion is zero.

7)  Errors are found in writing.
 7a) The "a series" in Line 20 should be "a series of ".
Reply:  Implemented. We have replaced "a series" with "a series of ".

7b) The "0.2"in Line 57 should be "0.2 mm/s".
Reply: Implemented. We have replaced "0.2" with "12 mm/min" in order to keep shear rates unit consistent in this manuscript.

7c) The "0.05" in Line 59 should be "0.05mm/min".
Reply: Implemented. We have replaced "0.05" with "0.05 mm/min".

 7d) The "a 5%"in Line 59 should be "5%".
Reply: Implemented. We have replaced "a 5%" with "5%".

7e) The "ring-shear tests" in Line 63 should be "ring shear tests".
Reply: Implemented. We have replaced "ring-shear tests" with "ring shear tests".

7f) The "0.02" in Line 65 should be "0.02 mm/min".

[Figure]

Reply: Implemented. We have replaced "0.02" with "0.02 mm/min".

7g) The "mud stone" in Line 66 should be "mudstone".
Reply: Implemented. We have replaced "mud stone" with "mudstone".

7h) The "range from" in Line 70 should be "ranging from".
Reply: Implemented. We have replaced "range from" with "ranging from".

7i) The "4.5" in Line 104 should be "4.5m".
Reply: Implemented. We have replaced "4.5" with "4.5 m".

7j) The "0.001" in Line 159 should be "0.001 degree".
Reply: Implemented. We have replaced "0.001" with "0.001 degree".

7k) The "drain" in Line 177 should be "drained".
Reply: Implemented. We have replaced "drain" with "drained".

7l) The "Figure 3b, 4b, and 5b" in Line 192 should be "Figures 3b, 4b, and 5b".
Reply: Implemented. We have replaced "Figure 3b, 4b, and 5b" with "Figures 3b, 4b, and 5b".

7m) The "as$\tau r/\sigma n$ (0.1)" and "and$\tau r/\sigma n$ (1)" in Line 251 should be "as $\tau r/\sigma n$ (0.1)" and "and $\tau r/\sigma n$(1)".
Reply: Implemented. We have corrected them as the reviewer's suggestion.

iscussion paper

7n) The "ÑDr(1) - Ñˇ      Dr(0.1)" throughout the manuscript should be italic.ˇ

Reply: Implemented.

8)    Errors are found in figures. In Figures 3, 4 and 5, the unit of normal stress should be "kN/m2". For example, in the Figure 4a, the unit of $\sigma$ should be "kN/m2" to keep consistent with the text body.

Reply: Implemented.

Interactive

9)    Apart from rate of shear, additional factors, such as slaking property and dynamic loading, also affect the residual strength of sliding surface of landslides, and in turn governs the stability of colluvial landslides. The following literature may be related to the additional factors and colluvial landsides, and may be cited, if applicable.

Reply: After reading the following references, we cited two references as suggested.  It is widely recognized that a reduction in shear strength of soils is closely associated with increasing of moisture content within soils (Shen et al., 2018; Tang et al., 2015; Yan et al., 2018), thus ring shear tests were conducted on saturated moisture content corresponding to the worst condition in engineering in this study.  See details on lines 116-120 of revised manuscript.

Li C, Yan J, Wu J, et al. Determination of the embedded length of stabilizing piles in colluvial landslides with upper hard and lower weak bedrock based on the deformation control principle. Bulletin of Engineering Geology and the Environment, 2017: 1-20.

Yan C, Xu X, & Huang L. Identifying the impact factors of the dynamic strength of mudded intercalations during cyclic loading. Advances in Civil Engineering, 2018.

Shen P, Tang H, Huang L, & Wang D. Experimental study of slaking properties of red-bed mudstones from the Three Gorges Reservoir area. Marine Georesources & Geotechnology (in the press).

[Figure]

iscussion paper

---

## Author Comment (AC2) · 18 Dec 2018

Review of manuscript nhess-2018-270 Title: "Influence of shearing rate on the residual strength characteristic of three landslides soils in loess area" by Baoqin et al.

Dear editor,

The paper presents the results of shear experiments conducted on 24 loess samples collected from three landslides. Samples were sheared in a rotary apparatus under two imposed velocities and four different normal stresses. The authors summarized the results and briefly discuss them. This manuscript should be better organized and written. Language mistakes make it sometimes impossible to read or understand. In addition, the authors should clarify what is innovative in this paper or how their results

support or differ from previous studies. I recommend the authors to significantly revise the manuscript before resubmission. Hope this helps

[Figure]

We thank for your constructive comments. The manuscript has been significantly improved by incorporating your suggestions. The following are our point-to-point responses to your comments.

Major comments

1. I found it hard to understand the connection between the definition of the residualstrength of a landslide and what is actually measured in the lab experiments. Some of my misunderstanding may cause by the fact that I'm not a soil mechanics expert. Saying that, I think that the authors should carefully define the basic terms and explain how they connect to their experiments. For example, the first sentence of the introduction says "Residual strength of soil is of great significance for evaluating the reactivating potential of the slope, in which consists of pre-existing sliding surface". I'm not sure how the current experiments deals with the "reactivating potential" as they suggest only one continuous sliding event in their experiments. In addition, how about the sliding surfaces? Did they form during the experiments?

Reply: Implemented. Residual shear strength can be used to evaluate the reactivate potential of the slope, in which consists of pre-existing sliding surface. Furthermore, it is also important for assessing stability for the slip surface of first-time natural or excavated slope failures (Mesri et al. 2005). We have check three landslides' history and found they are all original, so the sentence "residual shear strength can be used to evaluate the reactivate potential of slope…." is not appropriate to be used here.

We have revised this sentence as "residual shear strength of soil is of great significance for evaluating the stability for the slip surface of first-time natural or excavated slope failures" in revised manuscript. See details on lines 45-47 of revised manuscript.

[Figure]

2.    It seems that one important point of the experiments is the imposed shear velocity. The authors should clearly define in the introduction what are low and high velocities. I would rather use m/s as the velocity units, but in any case they should be consistent along the paper (see lines 52 and 57 as an example).

Reply: It is a nice suggestion.  With regards to define in the introduction what are low and high velocities: Following the previous study conducted by Tika et al. (1996), shearing rate higher than 10 mm/min is defined as high shearing rate, whereas, shearing rate lower than 10 mm/min is defined as low shearing rate. See detail on lines 65-66 of revised manuscript.

With regards to the velocity units: Following previous research (Bhat, 2013; Lupini et al., 1981; Sassa et al., 2004; Tiwari and Marui, 2005),  we used mm/min as shearing rate unit in this study. We have replaced "m/s" with "mm/min" in line 57 in order to keep shearing rate unit consistent along the paper.  See line 80 for detail.

3.    The motivation for this study is not clear to me. In lines 68-69 the authors suggest that not enough studies conducted on the issue that they just summarized in details in the two paragraphs above it. They should clarify what is the gap and how their new study contributes to the understanding of the problem.

Reply: Implemented. With regards to the gap:  On general, the effect of the shearing rate on the residual strength of the soil has not been sufficiently studied in high and

slow shearing rate range. Many studies above have been conducted on the residual shear strength of soils, and some inconsistent or even opposite results have been reported in the literature, which implied that there is still a lack of experimental data on this topic. Furthermore, research on the impact of the shearing rate on the residual strength of loess soil in relatively lower shearing rate range from 0.1 mm/min to 1 mm/min is scarce. See details on lines 96-102.

With regards to how the new studies contributed to the understanding of the problem: it should be noted that the residual strength parameters obtained from using different shearing rate may be adopted to provide a guide for designing some precision engineering which require high accuracy of the design parameters, thus, the effect of the shearing rate on the residual strength of soils should be fully understood to determine the parameters with high reliability. Accurate determination of the residual strength parameters and their dependence on the shearing rate may affect the stability evaluation of landslides. Thus, it is necessary to study the residual strength variation of loess in rate of shearing in order to have a good understanding of the suitable approach for the residual strength measurement. See details on lines 102-112 of the revised manuscript.

4. In the "Results and discussion" chapter many of the results are not discussed. For example, in sub-section 4.1 no discussion is following the observed difference between samples. Moreover, there are a lot of details with almost no discussion!

[Figure]

Reply: Implemented. We have added "Discussion" chapter in the revised manuscript, see details on lines 434-471 of the revised manuscript. Contents of Discussion are shown as follows:

Examination of the ring shear test results provides a basis for some general comments on the use of tests results with different shear rates, partially deepening some aspects deriving from previous studies.

From the experimental results on the three selected landslides, it was found that there is a negative relationship between residual friction coefficients and shear rates for all samples (Figs. 7, 8 and 9). Such a negative effect of shear rate (higher residual friction coefficients at lower rates) has been reported in the literature for fine-grained soils (Gratchev and Sassa, 2015; Tika et al., 1996a). This effect may be closely associated with ability of clay particles in specimen to restore broken bonds at different shear rates. Previous studies (Osipov et al., 1984; Perret et al., 1996). concluded that with higher shear rates, the breakdown of the bonds between clay particles or flocs exceeds the restoration bond, leading to reduction in residual friction coefficients. In contrast, the bonds between particles are rebuilt quickly and the recovery rate can catch up the breakdown rate at lower shear rates. Therefore, the weaker bonding between particles could explain the strength drop when the shear rate increases in this study.

[Figure]

The difference between the friction coefficients, $\tau_r/\sigma_n$ (1)-$\tau_r/\sigma_n$ (0.1), at each normal stress level varies in different locations. $\tau_r/\sigma_n$(1)-$\tau_r/\sigma_n$(0.1) in Ydg specimen are greater compared with that in Djg and in Dbz specimen (Table 2). As for Ydg and Dbz specimen, it is found that the shearing rate effect on the friction coefficient can be seen to decrease with increasing of normal stress (Figs. 7 and 8). However, the influence of shear rate on the friction coefficient increase with normal stress in Djg specimen (Fig. 6). Gibo et al. (1987) reported that the residual friction angle of soils was controlled by the effective normal stress as well as by the CF. Interestingly, Ydg (with CF 9%) and Dbz (with CF 9.1%) specimens with almost the same fraction of clay showed similar shear rate effect on the residual friction coefficient with normal stress increasing, however, Djg (with 24% CF) showed the contrast tendency of shearing rate effect on residual friction coefficient with normal stress, indicating that such effect is closely associated with CF. Therefore, as for Ydg and Dbz with relatively low fraction of CF, there is an increase effect of shear rate on residual friction coefficient with decreasing of normal stress. Thus, for the application of measured residual friction coefficient for stability analysis of shallow landslides with lower overburden pressure, it is significant for us to use a low shear rate in ring shear tests to measure residual shear strength parameters. On other hand, for Djg with high CF, it is more reliable to use a low shear rate in ring shear tests to determine residual friction coefficient for stability analysis of deep landslides with high overburden pressure.

[Figure]

5. I really recommend the authors to send the manuscript to English proofreading before resubmitting the manuscript.

Reply: Implemented. We have revised our manuscript with help of a native English speaker.

Moderate and minor comments

1. Figure 1 - Please increase line width of the arrows that point on "locations of study". I think it is better to add a small symbol for the location of each site. What are the red dashed lines in the photos? It should be described in the figure caption.

Reply: Implemented. With regards to line width: We have increased line width of the arrows that point on "Locations of study".

With regards to symbols for the locations of sites: we have added a small symbol (Jingyang) for the location of Dbz , and have added a symbol (Shaanbei) for the location of (Djg and Ydg).

With regards to red dashed lines in the photos: Red dashed lines in the photos means landslide boundary. See line 170 of the revised manuscript for detail.

[Figure]

2.	Line 127 – Why did you crush the material? Is it because the fragments are too big for the cell? Please explain in the text.

Reply: Implemented. We have explained the reason in the text as follows: After soil samples were air-dried, small lumps may exist in samples which may be too big for the cell, so we crush lumps in order to make the soil uniform. This should be done with care so as not to destroy silty-dominated loess. See details on lines 182-186 of the revised manuscript.

3.	Line 128 - What is the desired content? Please refer to a table and/or give the numbers here.

Reply: Implemented. We have replaced "desired content" with "saturated water content". See details on line 187 of the revised manuscript.

4.	Line 131 - When during the procedure you sieved the material? Before or after adding the distilled water? Before or after crushing?

Reply: After crushing, we sieved the material. Then we added distilled water to the sieved soils.  See lines 182-186 of the revised manuscript for detail.

5.	Line 132 – "physical indexes". Do you mean the physical parameters that are listed in the line above?

Reply: Yes, with regards to "physical indexes", we mean the physical parameters listed in Table 1

6. Table 1 – Check units! Replace m with mm.

Reply: Implemented. See details on Table 1.

7. Lines 147-148 - It is not sounds to me like the most important thing about ring shear test apparatus, as direct shear is better in that sense. I would say that unlimited displacement is the most important advantage of the rotary shear for this specific application.

Reply: It is a nice suggestion. The most important advantage of ring shear apparatus is its ability to shear unlimited displacement. As suggested by the reviewer, we have modified this sentence as follows: The great advantage of a ring shear apparatus to measure residual shear strength is its ability to allow unlimited shear displacement. See details on lines 53-56 of the revised manuscript.

8. Line 155 – "single direction", what do you mean? It is a rotary shear and not direct shear apparatus!

Reply: We conducted ring shear tests in single direction in this study since the residual state was attained in single direction. As for ring shear apparatus, we will modify this sentence as follows: The apparatus is capable of shearing the specimen for large displacement. See details on lines 221-223 of the revised manuscript.

9. Lines 155-157 and 178 – The annular porous plates, explain why they are needed! I recommend including an image that shows them.

[Figure]

Reply: Implemented. With regards to the reason why the annular porous plates are needed: Two annual porous plates were used to provide drainage condition in the test following previous research (Stark and Vettel, 1992). See details on lines 227-228 of revised manuscript.

With regards to an image that shows porous plates: We have added an image. See Figure 2 in the revised manuscript as follows:

[Figure]

Figure 2. Ring shear apparatus (SRS-150)

[Figure]

10. Line 158 – What is the sampling rate of the rheology data?

Reply: Implemented. We have added the sampling rate as follows: shear strength of loess specimen was recorded at intervals of 1s before the peak shear strength, after the peak, the sampling rate was increased to 1min. See details on lines 255-256 of revised manuscript.

11. Figure 2 - Please change the font color and add a background to improve visibility.
I would highly recommend adding a sub-figure here that shows a cross section of the shear box and clarify where the actual soil sample is, and which part is rotating and which is stationary.

Reply: With regards to the font color: We have tried to use white font, green font and blue font etc., unfortunately we found it is not clear compared with using red font, thus we keep font color here.

With regards to a sub-figure here: We have added a sub-figure here to show cross section of the shear box. Figure 2 was shown as follows in revised manuscript.

[Figure]

[Figure]

12. Line 172 – What is the required consolidation? How do you know if you get it or not?

Reply: Implemented. Consolidation was completed when the consolidation deformation was smaller than 0.01 mm within 24 hr (Kramer et al., 1999; Shinohara and Golman, 2002). Please see details on lines 250-251 of the revised manuscript.

[Figure]

comment

13. I would combine sections 3.1 and 3.3 or better separate the text.

Reply: Thanks for your suggestion, but we decided to separate testing sample and testing procedure after consideration.

14. Line 195 – The authors suggest they ran the samples for large displacements. Did they run at least one experiment beyond the maximum displacement given in the figures to ensure that they really get to a steady state friction.

Reply: Yes, we have conducted experiments beyond the maximum displacement given in the figures and we found it is appropriate to shear samples until the magnitude of displacements attaining values shown in the manuscript. Furthermore, we following the (Bromhead, 1992), the residual stage is attained if a constant shear stress is measured for more than half an hour. See details on lines 289-290 of the revised manuscript.

15. Lines 222-224 - You should better explain what actually this reorientation is and why do you think it should be different specifically for Djg samples?

Reply: Skempton (1985) reported that the strength of soils falls to the residual value in ring shear tests, owing to reorientation of platy clay minerals parallel to the direction of shearing. Based on the conclusion that the post-peak drop in strength of normally consolidated soil is only due to particle reorientation after the peak strength (Mesri and Shahien, 2003; Skepmton, 1964), the results in this study demonstrated that Djg samples show the greatest strength drop in comparison with Ydg samples and Dbz samples (Figs. 3-5), thus, we think

[Figure]

Djg samples show greatest particle reorientation. See details on lines 322-328 of the revised manuscript.

16. Line 296 - I'm not sure what the connection between these soil properties is. Please define them and better connect to the measurements.

Reply: Implemented. With regards to the connection between residual shear strength and soil properties: It has been recognized that residual shear strength of soils closely related with soil properties, such as particle size distribution, liquid limit, plasticity index and clay fraction (Terzaghi et al., 1996). See details on lines 417-419 of the revised manuscript.

17. The current Conclusions chapter is mostly a summary of the results, instead of a short description of what we have studied and can take from this experimental study.

Reply: We have added "Discussion" chapter in the revised manuscript, see response to Comment # 4 of this reviewer.
* * *